# The Most Valuable Predictive Factors for Bronchopulmonary Dysplasia in Very Preterm Infants

**DOI:** 10.3390/children10081373

**Published:** 2023-08-11

**Authors:** Wenwen Chen, Zhenhai Zhang, Liping Xu, Chao Chen

**Affiliations:** 1Zhangzhou Municipal Hospital of Fujian Province and Zhangzhou Affiliated Hospital, Fujian Medical University, Zhangzhou 363000, China; 19111240024@fudan.edu.cn (W.C.); zhenhai0596@126.com (Z.Z.); 2Children’s Hospital, Fudan University, Shanghai 201102, China

**Keywords:** BPD, preterm infants, risk factors, chest radiograph

## Abstract

Introduction: It is urgent to make a rapid screening of infants at the highest risk for bronchopulmonary dysplasia (BPD) via some succinct postnatal biomarkers, such as *Ureaplasma Urealyticum* (*UU*) infection and chest radiograph images. Methods: A retrospective study was performed. Moderate to severe BPD or death was set as the main outcome. The association between putative variables and the main outcome were assessed by bivariate analyses and logistic regression. Results: A total of 134 infants were enrolled. Bivariate analyses showed the gestational age, birth weight, appearances of diffuse opacities or grid shadows/interstitial opacities or mass opacities or cystic lucencies on chest radiographic images, a ductal diameter ≥1.5 mm and whether *UU* infection was associated with BPD. After adjustment by logistic regression, the risk of BPD with gestational age, sex and specific chest-radiographic manifestations remained significant. Conclusions: Chest radiograph images (appearance of diffuse opacities or grid shadows/interstitial opacities or mass opacities or cystic lucencies) could provide a quick prediction of developing BPD in clinical practice, in addition to gestational age and sex. *UU* infection was not an independent risk factor for BPD.

## 1. Introduction

Bronchopulmonary dysplasia (BPD) is a common chronic lung disease in preterm infants that is associated with long-term lung dysfunction, cardiovascular involvement, extrauterine growth retardation and neurodevelopmental disorders [1]. In essence, it is a developmental disorder of immature lungs caused by various harmful insults, such as hyperoxia, mechanical ventilation and infection. In the past two decades, despite the improvement of perinatal medicine, the incidence of BPD has remained stable, accounting for about 32–45% of preterm infants born at 22–28 weeks gestational age based on the need for supplemental oxygen at 36 weeks postmenstrual age (PMA) [2,3]. The unchanged BPD rates might probably be due to the improved survival of extremely preterm infants who are born at an earlier stage of lung development that is more vulnerable to insults [4], highlighting the continuing challenges for clinicians facing these infants at the highest risk. Since relatively few of the therapeutic approaches have been supported by high-quality evidence, the concern for BPD should begin from the very beginning with a risk-factor identification, which will enable appropriate strategies for prevention and mitigation at an early stage.

Several factors have been suggested to predict the risk for BPD. It is generally accepted that susceptibility to BPD is partially determined by nonmodified factors (gestational age, birth weight and sex) and intrauterine exposure, such as maternal smoking and hypertension, fetal growth restriction (FGR), and chorioamnionitis [5]. Furthermore, various postnatal insults, such as mechanical ventilation, patent ductus arteriosus (PDA), infection, and prolonged parenteral nutrition, might also contribute to BPD [6]. However, these multiple factors have sometimes caused overloaded information and made it difficult to form interpretations. Some factors only had a low predictive value for BPD, while some factors were overestimated, being confounded by other factors. The search for more objective indicators that have the potential to differentiate subjects with the highest risk for BPD remains a strong attraction. Thus, novel imaging, biochemical factors, and molecular factors have been proposed and studied with clinical interest in order to find better alternatives.

A chest radiograph is the most convenient tool for evaluating respiratory problems in seriously ill infants. In 1967, Northway et al. first reported four stages of chest-radiographic findings that characterized classical BPD [7], which could be explained by corresponding histological findings. The four stages reflected the progression of lung injury. In brief, at first, chest radiographs usually presented homogeneous lesions, such as a diffuse fine granularity or opacities, and then turned into heterogeneous images, such as lucencies mixed with density, bubbly-cystic appearances, or irregularly dense strands in the chronic phase. However, this evolution of chest radiographs became indistinct or overlapping due to the introduction of antenatal steroids, pulmonary surfactants and lung-protective ventilator strategies brought by the advances of perinatal and neonatal care. As a result, the typical bubbly-cystic appearance on chest radiographs frequently appeared earlier in infants born at more advanced gestational ages. Even before that time, a coarse interstitial thickening pattern could yield a high specificity for predicting BPD [8]. In addition, a postnatal morbid state, such as PDA and pneumonia, which were considered to be associated with BPD, could also cause new changes on chest radiographs. In view of the above, chest radiograph images could yield meaningful information about lung lesions and logically serve as a biomarker for BPD.

*Ureaplasma urealyticum* (*UU*) is an opportunistic pathogen that colonizes the urogenital tract of women [9]. It had a highly detectable rate in pregnant women who suffered from a premature rupture of membrane (PROM) [10]. Infants delivered by these women usually carried *UU* on their respiratory tract and were defined as being at risk of developing BPD [11]. However, this point of view was challenged by some researchers, since they failed to demonstrate an association between positive *UU* screening and BPD [12]. The controversial results may be attributed to the different subjects and methods of analysis.

Based on previous studies, we designed this study with the aim of investigating: (1) the association between *UU* infection and the development of BPD in a cohort of very preterm infants; and (2) the value of chest-radiographic images in predicting BPD.

## 2. Methods

### 2.1. Subjects and Data Collection

This was a retrospective study. Preterm infants born at less than 32 weeks gestational age and surviving to 36 weeks’ PMA admitted to the neonatal intensive care unit (NICU) of Zhangzhou Affiliated Hospital of Fujian Medical University between January 2022 and December 2022 were included in this study. Infants with major congenital malformations and those who were forced to withdraw treatment by parents were excluded. Clinical data were collected from the hospital’s electronic medical record system. This study was approved by the Institutional Review Board (IRB) of Zhangzhou Affiliated Hospital of Fujian Medical University, with a waiver of informed consent (2022KYZ297).

### 2.2. Definitions

Moderate or severe BPD or death was set as the main outcome because this degree of severity is highly associated with long-term adverse outcomes of preterm infants. Diagnosis of BPD was based on the NICHD Workshop summary [13]. Preterm birth was defined as born at less than 37 gestational weeks. Very preterm birth was defined as born at less than 32 gestational weeks. Extremely preterm birth was defined as born at less than 28 gestational weeks. Moderate BPD was defined as oxygen supplementation for at least 28 days and persistent need for oxygen (FiO_2_ < 30%) at 36 weeks PMA. Severe BPD was defined as oxygen supplementation for at least 28 days and persistent need for oxygen (FiO_2_ ≥ 30%) and/or ventilatory support (mechanical ventilation or positive pressure) at 36 weeks postmenstrual age.

In our NICU, infants had a chest radiograph at their bedside within the first hours of life, and subsequent chest radiographs were performed according to the condition. Chest radiographs obtained within 14 days of life were reviewed for analysis. Four types of images were supposed to be meaningful: (1) diffuse opacities (blurred cardiac and diaphragmatic edge or white lung) on two consecutive chest radiographs within 7 days of life; (2) grid shadows/interstitial opacities; (3) mass opacities; (4) cystic lucencies (Shown in Figure 1). Any one of the above criteria was considered positive.

Color Doppler echocardiography was first performed around the third day of life in every preterm infant and rechecked if necessary. PDA was confirmed if ductal flow was visualized, irrespective of hemodynamic significance, and the calibers were measured. A ductal diameter ≥ 1.5 mm was considered positive.

*UU* examination was performed in every preterm infant via testing nasopharyngeal secretion or tracheal aspirates by PCR at admission. *UU* colonization was defined as a PCR test positive for *UU*. *UU* infection was diagnosed if both of the following were satisfied: (1) *UU* colonization; (2) two consecutive elevations of total white blood counts (both ≥ 20 × 10^9^/L) within 7 days of life, which suggested an inflammatory response caused by *UU* [14].

Serum CRP was also included in analysis because it was a frequently used monitor of infection in NICU. Generally, the first CRP was acquired after 6 h of birth and monitored during 3 consecutive days. Two consecutive elevations of CRP (both ≥ 10 mg/dl) within 7 days of life were considered to be an intrauterine infection.

### 2.3. Statistical Analysis

Analysis was performed via SPSS version 26 software. Descriptive data were presented as means ± SD or percentages. To compare differences between two groups, chi-square tests for categorical variables and two-sample t-tests or Mann–Whitney U tests for continuous variables were applied. Analyses of the main outcome and risk factors were performed via logistic regression, with 95% CIs calculated for ORs adopting the forward entry method. A *p*-value of ≤0.2 was used in univariate analysis for inclusion of putative risk factors into the multivariate (adjusted) model. A *p*-value of <0.05 was considered to be statistically significant.

## 3. Results

### 3.1. Clinical Characteristics of the Subjects

During the study period, 149 infants were born before 32 gestational weeks and admitted to our NICU. Among them, 15 infants were excluded: 13 infants who died before 28 days of life and 2 infants who had a tracheoesophageal fistula malformation. In total, 134 infants were included in the analyses. The mean gestational age was 29.5 ± 1.7 weeks (range 24^+5^~31^+6^ weeks), and the mean birth weight was 1335 ± 313 g (range 600~2100 g). The male-to-female ratio was 1.6:1 (83 males and 51 females). Seventeen infants were diagnosed as moderate to severe BPD, and four infants died. The dead infants were all born extremely preterm (one 25 gestational weeks and three 27 gestational weeks), with a birth weight under 1000 g. The incidence of moderate to severe BPD or death in the studied subjects was 15.7% (21/134). The *UU* colonization rate was 22.4% (30/134), and the incidence of *UU* infection was 8.2% (11/134).

### 3.2. Factors Associated with Moderate to Severe BPD or Death

Bivariate analyses showed that a lower gestational age and birth weight, PDA, *UU* infection, and appearances of diffuse opacities or grid shadows/interstitial opacities or mass opacities or cystic lucencies on chest-radiographic images were factors significantly associated with moderate to severe BPD or death before 36 weeks PMA (Table 1). Multivariate analysis with these significant associated factors included in the logistic regression model revealed that a lower gestational age, male sex, and appearances of diffuse opacities or grid shadows/interstitial opacities or mass opacities or cystic lucencies on the chest radiograph were independently associated with moderate to severe BPD or death before 36 weeks PMA. The adjusted ORs and sensitivity, specificity, positive predictive value and negative predictive value are shown in Table 2 and Table 3. After adjustment by gestational age and sex, PDA and *UU* infection were not significantly associated with BPD. Among the dead infants, three displayed mass opacities, and one displayed diffuse grid shadows in chest radiograph images. All of them had a PDA with a diameter between 2–3 mm, and only one of them had *UU* colonization.

## 4. Discussion

The aim of study was to screen out several valuable predictors for quickly identifying infants who might develop BPD in clinical practice. Although previous studies have already accessed and recommended risk factors for BPD, the numerous factors usually confused clinicians and made them search for a more intuitive approach with fewer independent factors.

Bivariate analyses suggested that PDA and *UU* infection might have potential impact on BPD, but this concern was dismissed after adjustment by gestation age and sex. Unadjusted ORs also have a certain relevance because they combine the “direct effect” resulting from the putative factor (PDA or *UU* infection) itself and an “indirect effect” more greatly caused by a low gestational age [15]. The interpretation of these findings might lie in the fact that PDA or *UU* infection contributed indirectly to BPD through gestational age. Although the *UU* colonization rate was higher in BPD infants, after adjustment for gestational age and sex, *UU* infection was not significantly associated with BPD. The problem of identifying infants with PDA or infected by *UU* with the highest risk for developing BPD still needs to be solved. As was shown in our study, the remaining significant factors for BPD were a lower gestational age, male sex and appearances of diffuse opacities or grid shadows/interstitial opacities or mass opacities or cystic lucencies on a chest radiograph. In summary, the most valuable predictors for BPD were gestational age and a chest radiograph.

Maturity at birth is the strongest predictor of not only survival, but also complications in preterm infants [16]. The overall incidence of BPD increased, with a decrease in the gestational age, and those who were born extremely preterm had the highest risk for BPD [17]. It is widely accepted that the key points of the pathogenesis of BPD are an immature lung encountering oxidative stress, which generates an inflammatory cascade. Infants born at less than 32 weeks are often exposed to supplemental oxygen in the late canalicular or saccular phases of lung development, and the subsequent morphological development might be interrupted by postnatal insults. Compared to the hypoxia environment in utero, even exposure to routine room air imposes higher oxygen to the immature lung. Since gestational age biologically determines the critical period in lung development, it is the dominant predictor of BPD and takes on a “dose-response” relation [18]. Apart from that, gestational age was considered to be a confounder variable in the causal pathway between putative risk factors and neonatal outcomes. Customarily, adjusting for gestational age was applied in observational studies for analysis. In addition to gestational age, birth weight is another strong predictor for adverse outcomes in preterm infants [18]. However, there is a perfect collinearity between gestational age and body weight, which might greatly affect the effect of other factors in logistic regression. Given this, most studies only retained gestational age for adjustment to assess the independent effect of other factors in logistic regression [15], which was also the method we chose to analyze the data in this study.

After adjustment for gestational age, male sex still appeared to increase the odds for BPD, and this finding is consistent with other studies [19]. The sex differences have also been found in other neonatal outcomes [20]. However, the underlying mechanisms remain unclear. In the early 1980s, Torday JS et al. demonstrated that fetal pulmonary maturity was higher in females than in males [21]. Recently, Lien YC et al. identified transcriptome alterations in placentas, involving energy metabolism, mitochondrial function, inflammation, and detoxification, with fetal sex disparities [22]. Some scholars speculated that the growth of airways in males lagged behind lung parenchyma and thus resulted in a smaller airway caliber and lung size relative to females [23].

The chest-radiographic images serving as a postnatal predictive factor in our study were easy to identify and achieved a good value. Cystic lucencies, to some extent representative of alveolar growth abnormalities, were recognized as the typical radiographic pattern of BPD. However, this typical appearance was less frequent within the first days of life. As supplementation, we incorporated radiographic signs of diffuse opacities (blurred cardiac and diaphragmatic edge or white lung), grid shadows/interstitial opacities and mass opacities, which could be present ahead of cystic lucencies. With the prevalence of antenatal steroids clinical practice, the incidence and severity of respiratory distress syndrome (RDS) has decreased [24]. An appearance of near or definite white lung on a chest radiograph taken within the first hours of life was correlated with severe RDS or might indicate acute respiratory distress syndrome (ARDS) that did not respond well to single-dose surfactant replacement. ARDS is a common cause of respiratory failure in critically ill patients, resulting in increased morbidity and mortality. Laboratory studies have demonstrated inflammation response resulting in both alveolar epithelial and endothelial injury in ARDS [25]. Infants with ARDS depended on invasive ventilator-assisted and iNO treatments [26]. Considering the physiopathologic mechanism and ventilation mode, ARDS seemed to impose a potential hazard on BPD. Unfortunately, none of the studies assessed the effect of ARDS as an independent risk factor on BPD, probable due to the dilemma of diagnosing ARDS in preterm infants in a definite way. Although a consensus on neonatal ARDS was proposed in 2017, it is still very difficult to distinguish RDS from ARDS within the first three days after birth because RDS could also be a trigger of ARDS, and the time for the diagnosis of ARDS and RDS often overlaps. Empirically, an advancing demand for ventilation parameters is a warning of ARDS. Objectively, a series of chest radiographs could provide meaningful signs for a coarse differentiation. Pure RDS recovered quickly with a clearance on a repetitive chest radiograph, while ARDS remained the same or with aggravated opacities. Grid shadows/interstitial opacities always indicated interstitial lesions, including airway, alveolar wall and pulmonary vascular damages and interstitial edema. Previous studies have demonstrated interstitial changes on a chest radiograph to be a predictor for BPD [27]. In addition, interstitial changes were less influenced by inflation status and were easily noticeable and present earlier than cystic lucencies on chest radiograph. Mass opacities were considered as symbols of lung consolidation, representing an infectious process that caused the alveoli, or air sacs, to fill up with fluid or pus. Infants born very preterm and those who had impaired swallowing mechanisms were at a high risk of infection. The overwhelming inflammation during the infectious process brought devastating damage to the developing lung [28]. Bilateral opacities distributed in perihilar regions with an enlarged cardiac silhouette were common signs of pulmonary edema, induced by hemodynamically significant PDA [29]. The overload of pulmonary fluid can affect lung compliance, gas exchange, pulmonary vasculature and myocardial function, thus increasing the need and duration of mechanical ventilation and oxygen needs [30]. These images all have a pathophysiological basis relevant to lung injuries, as a result of which the interpretation of their predictive values makes sense. This clinic-radiological correlation can help in early diagnosis and prognosis of BPD after birth. In addition, a specific radiograph appearance can also provide a reference for treatment. For example, pulmonary edema might suggest the need for fluid restriction, diuretics application and PDA closure, and cystic lucencies always suggests a higher tidal volume requirement.

There were several limitations to our study. First, it was from a single center that could not avoid selection bias. Second, the retrospective nature of the study does not allow direct relational associations to be extracted, which should be more clearly expressed. Third, the small sample size did not allow further analysis at each gestational age, respectively.

The strength of our study was that we provided a quick and simple way for clinicians to identify infants with the highest risk of BPD via specific features (gestational age, sex and chest radiograph images). The images we suggested as being positive (appearance of diffuse opacities or grid shadows/interstitial opacities or mass opacities or cystic lucencies) on the chest radiograph were easy to identify, which may avoid ambiguity and inconformity between observers.

## 5. Conclusions

Chest radiograph images (appearance of diffuse opacities or grid shadows/interstitial opacities or mass opacities or cystic lucencies) could provide a quick prediction of developing BPD in clinical practice, in addition to gestational age and sex. *UU* infection was not an independent risk factor for BPD.

## Figures and Tables

**Figure 1 children-10-01373-f001:**
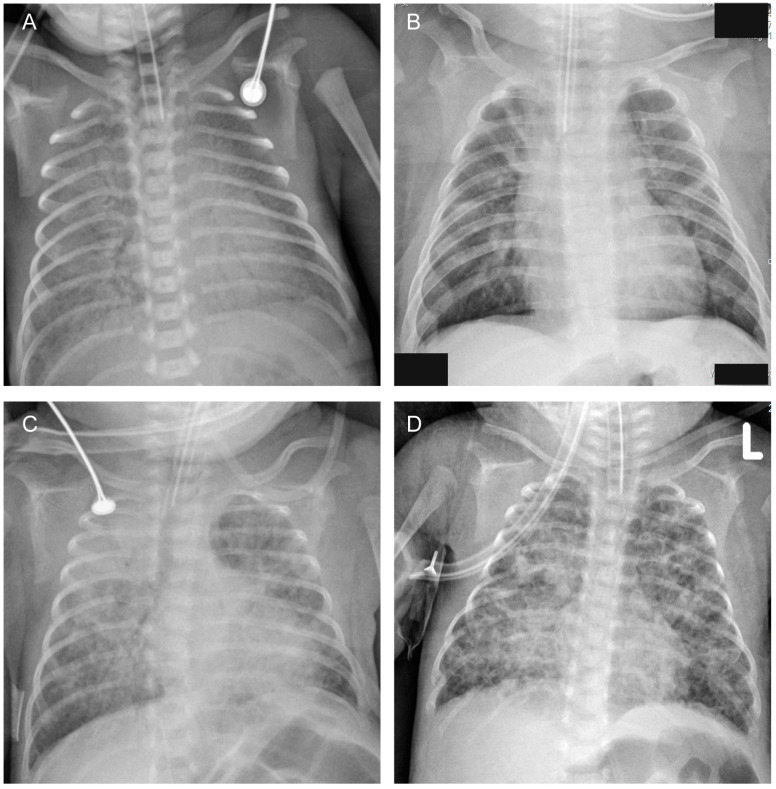
Chest radiographic images: (**A**) Diffuse opacities (blurred cardiac and diaphragmatic edge). (**B**) Grid shadows/interstitial opacities. (**C**) Mass opacities. (**D**) Cystic lucencies.

**Table 1 children-10-01373-t001:** Demographics of infants with moderate to severe BPD or death before 36 weeks.

Variables	No BPD or Death n = 113	BPD or Death n = 21	*t* or χ^2^	*p*
Gestational age (weeks) mean ± SD	29.5 ± 1.4	27.2 ± 2.0	5.009	<0.001
Weight (g) mean ± SD	1402 ± 275	976 ± 260	6.581	<0.001
Male (%)	66 (58.4)	17 (81.0)	3.818	0.051
*UU* colonization (%)	22 (19.5)	8 (38.1)	2.545	0.111
*UU* infection (%)	6 (5.3)	5 (23.8)	5.776	0.016
PDA (%)	51 (45.1)	16 (76.2)	6.833	0.009
PDA ≥ 1.5 mm (%)	43 (38.1)	14 (66.7)	10.829	0.004
Chest radiographs: diffuse opacities or grid shadows/interstitial opacities or mass opacities or cystic lucencies (%)	13 (11.5)	12 (57.1)	21.391	<0.001
Serum CRP ≥ 10 mg/dl (%)	12 (10.6)	4 (19.0)	0.529	0.467

BPD—Bronchopulmonary dysplasia, *UU*—*Ureaplasma Urealyticum*, PDA—Patent Ductus Arteriosus, CRP—C reactive protein, PMA—postmenstrual age.

**Table 2 children-10-01373-t002:** Multivariate analysis with adjusted odds ratios for BPD or death before 36 weeks PMA.

Factors	OR	95%CI		*p*
		Upper Limit	Lower Limit	
Gestational age (divided into 3 groups: <28 weeks, 28~29^+6^ weeks, 30~31^+6^ weeks)	4.6	2.1	10.1	<0.001
Chest radiographs: diffuse opacities or grid shadows/interstitial opacities or mass opacities or cystic lucencies	6.8	2	23.1	0.002
Male sex	3.9	1	15	0.049
*UU* infection	-	-	-	0.288
PDA ≥ 1.5 mm	-	-	-	0.118

BPD—Bronchopulmonary dysplasia, *UU*—*Ureaplasma Urealyticum*, PDA—Patent Ductus Arteriosus, PMA—postmenstrual age.

**Table 3 children-10-01373-t003:** Predictive characteristics of associated factors for moderate to severe BPD or death before 36 weeks PMA.

Variables	Cut-Off Value	Sensitivity	Specificity	PPV	NPV
Gestational age	28 weeks	67%	90%	56%	94%
	30 weeks	81%	60%	27%	95%
Male	-	81%	42%	20%	92%
Chest radiographs: diffuse opacities or grid shadows/interstitial opacities or mass opacities or cystic lucencies	-	57%	88%	48%	92%

BPD—Bronchopulmonary dysplasia, PMA—postmenstrual age, PPV—positive predictive value, NPV—negative predictive value.

## Data Availability

Not applicable.

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
