# Peer review of "The Most Valuable Predictive Factors for Bronchopulmonary Dysplasia in Very Preterm Infants"

_children, 2023, doi:10.3390/children10081373_

Round 1

Reviewer 1 Report

I read with interest the paper by Chen et al about the possible predictive factors for BPD.

The paper explores rediographic, microbiological and clinical data as possible predictors of the disease and concludes that only the gestational age, sex and a set of chest radiographic manifestation were found to be  independently predictive of BPD on multivariate analysis.

My first concern is about the scientific contribution of this paper, that obtains known results, except for the Ureaplasma Urealiticum infection, that however is not always a confirmed risk factor.

The abstract starts and ends talking about UU, but the title of the paper is a wider concept of "predictive factors". On the other side the radiographic appearance is not considered in the abstract introduction, but it is described as the primary outcome of the study.

The primary outcome as described is: "the most valuable chest radiographic pattern for predicting BPD under the current perinatal practices." But the authors never explore the different patterns in a separate way in the results.

Consequently I suggest to change the abstract and the primary outcome. 

Moreover, these patterns are frequently not exclusive, but mixed in the CXR of this population, consequently I would find it difficult to differentiate the single patterns.

The presence in the population of only 17 subjects with msBPD is on one side the signal of the selection of a low risk population, the other makes the analysis non so reliable.

Table 1: the described percentages are on the total of the row (eg. 22 UU colonizations in the noBPD group and 8 in the BPD group, the percentages are described as 22/30 and 8/30. It would be more important and interesting to describe the prevalence of every aspect on the total of noBPD and BPD subjects (e.g. no BPD UU colonizations: 22/113 (19.4%) and BPD UU colonizations: 8/21(38%)). This for every row.

Reviewer 2 Report

Thank you for the opportunity to review this article, which describes a retrospective study in which the authors analyze factors that may help predict the occurrence of pulmonary broncho-dysplasia in very preterm infants. I found the article quite interesting. Although it does not provide findings that can be considered novel, it does help to consolidate the importance of radiological imaging. It contributes by providing negative data on the involvement of Ureaplasma Urealyticum. Nevertheless, some minor aspects could be reviewed. These are as follows:

The article's length is perfect, as it is short and focuses on what is essential. However, in the introduction, I suggest that the authors add a line (at most two) justifying their work, linking the end of the text to their objectives.

In methods, the authors should first describe the type of study performed.

In the ethical aspects, the authors report that using informed consent was unnecessary. This statement is expected, as it is a retrospective study that only uses data from patient records. However, at the end of the article, the authors state that informed consent was obtained from the subjects. These two statements are a contradiction. The statement described in the body of the article is probably the correct one, but the authors must clarify it. In addition, the study subjects were preterm infants. If there was consent, their parents or legal guardians gave it, so the text at the end would not be correct either.

At the beginning of the discussion, the authors should state the main objective of their study. In addition, it sounds a bit strange to start talking directly about the incidence of severe BPD or death when there are more general data in their results that can serve as an introduction to the more specific aspects, thus making the development of the discussion a bit more natural and therefore easier to read.

On limitations and strengths, the authors should show the limitations first. There are a few more limitations than they refer to. For example, selection bias is unavoidable, mainly when the study is limited to a single center and one year. The retrospective nature is another significant limitation, as it does not allow direct relational associations to be extracted, which should be more clearly expressed. As for strengths, the study undoubtedly has them. However, choosing a cohort of newborns aged one year and from a single center cannot be described as a strength that reduces selection bias—quite the contrary. Second, the authors refer that the historical data were intact, which makes their analysis more reliable. They should define what they consider «intact» since conducting a study with poorly preserved data is difficult or impossible.

Using intact data should not be a study's strength but a minimum quality required for any investigation. Third, the fact that the radiographic images were easy to identify is not a strength either since there may have been bias in their interpretation, and that would be a limitation rather than a strength.

The conclusions could also be strengthened as they merely describe the key findings. I encourage the authors to draw absolute conclusions from their work, the implication this has from a practical point of view, and the opportunity they open up in terms of research. Also, it is curious that the first line of the abstract and the objectives of the article mention Ureaplasma Urealyticum infection, yet nothing is said about it in the conclusions. The conclusions must reflect the ultimate sense of why the research was conducted. Since the conclusions must go beyond the simple results of the study, it is essential to have described the study's limitations well. The limitations and conclusions sections are the ones that require the most revision on the part of the authors.

Reviewer 3 Report

Thank you for submitting your manuscript entitled “Assessment of Predictive Factors for Bronchopulmonary Dysplasia in Very Preterm Infants”. I have carefully reviewed the manuscript. I want to provide you with feedback and recommendations for improvement. 

Please find below my comments.

The study design should be reexamined to ensure its appropriateness and validity in addressing the research question. In the “Methods”, the authors have to explain why they take into consideration these specific risk factors as it is well known that many others, such as preeclampsia, chorioamnionitis, IUGR, infections, nutrition, ventilation mode and its related consequences such as barotrauma and volutrauma are also related to BPD. In addition, presenting information concerning the old and new BPD in the cases would be more beneficial. 

·      Line 27: Defining preterm and extremely preterm would be more appropriate.

·      Line 29-30: The authors have to discuss their state, “The reported rate of BPD has not changed over the past few decades, possibly due to increased survival of extremely preterm infants, highlighting the continuing challenges for clinicians facing these infants at highest risk”.

·      Line 48-50: Give more information about the radiographic images regarding overlap with other clinical entities. 

·      Line 50-51: Please clarify.

·      Line 57: Please use a transitional sentence. For example, you can rearrange the sentence “Beeton ML et al. found that presence of UU was significantly associated with the development of BPD, while Glaser K et al. failed to demonstrate an association between positive UU screening and BPD”.

·      Line 68: It would be nice to complete the sentence explaining that this biomarker could be used for early BPD screening.

·      Line 97-100: The authors have to clarify under which circumstances a color Doppler echocardiography was conducted.

·      Line 101-104: It would be helpful to clarify when and how UU colonization was examined. The authors should also present information about the number of infants that underwent tracheal aspiration. Moreover, the need for ventilation support, invasive or non-invasive, should be mentioned.
Please explain if 1) and 2) should be present simultaneously for a positive result.

·      Line 105-107: Explain under which circumstances CRP was measured.

·      Line 124:   Mention the exams performed and their results on the dead infants.

·      Figure 1: Please replace “A” with “A)” or “A.” etc

·      Line 150-152: Study limitations can be presented as a distinct paragraph before the conclusion.

·      Line 244-246: Study limitations can be presented as a distinct paragraph before the conclusion.

·      The authors could also write a paragraph explaining the benefit of this study in neonatology. 

These suggestions will help you improve your manuscript. 

Minor editing of English Language are required.

Round 2

Reviewer 1 Report

The authors adressed my comments adequately.

Reviewer 3 Report

The authors submitted their manuscript entitled “Assessment of Predictive Factors for Bronchopulmonary Dysplasia in Very Preterm Infants” related to the predictive factors for BPD. Although some information is already known, they aimed to provide a quick and simple way for clinicians to identify infants with the highest risk for BPD via specific features. 

The manuscript is well written. Therefore, I recommend the acceptance in the present form.